# A Novel Recessive Mutation in SPEG Causes Early Onset Dilated Cardiomyopathy

**Aviva Levitas[1], Emad Muhammad[2,3], Yuan Zhang[4,5], Isaac Perea Gil[4,5], Ricardo Serrano[6], Nashielli Diaz[4,5], Maram Arafat[2,3], Alexandra A. Gavidia[4,5], Michael S. Kapiloff[7], Mark Mercola[6], Yoram Etzion[8,9], Ruti Parvari[2,3], Ioannis Karakikes[4,5]***

1 Department of Pediatric Cardiology, Soroka University Medical Center and Faculty of Health Sciences, Ben-Gurion University of the Negev, Beer-Sheva, Israel, 2 The Shraga Segal Department of Microbiology, Immunology & Genetics, Faculty of Health Sciences, Ben-Gurion University of the Negev, Beer-Sheva, Israel, 3 The National Institute for Biotechnology in the Negev, Ben-Gurion University of the Negev, Beer-Sheva, Israel, 4 Department of Cardiothoracic Surgery, Stanford University School of Medicine, Stanford, California, United States of America, 5 Cardiovascular Institute, Stanford University School of Medicine, Stanford, California, United States of America, 6 Cardiovascular Institute and Department of Medicine, Stanford University, Stanford, CA, USA, 7 Cardiovascular Institute and Departments of Ophthalmology and Medicine, Stanford University, Stanford, CA, USA, 8 Regenerative Medicine & Stem Cell Research Center, Ben-Gurion University of the Negev, Beer-Sheva, Israel, 9 Department of Physiology and Cell Biology, Faculty of Health Sciences, Ben-Gurion University of the Negev, Beer-Sheva, Israel

* ioannis1@stanford.edu.

**Data Availability Statement:** All relevant data are within the manuscript and its Supporting Information files

## Abstract

Dilated cardiomyopathy (DCM) is a common cause of heart failure and sudden cardiac death. It has been estimated that up to half of DCM cases are hereditary. Mutations in more than 50 genes, primarily autosomal dominant, have been reported. Although rare, recessive mutations are thought to contribute considerably to DCM, especially in young children. Here we identified a novel recessive mutation in the striated muscle enriched protein kinase (*SPEG*, p. E1680K) gene in a family with nonsyndromic, early onset DCM. To ascertain the pathogenicity of this mutation, we generated *SPEG* E1680K homozygous mutant human induced pluripotent stem cell derived cardiomyocytes (iPSC-CMs) using CRISPR/Cas9-mediated genome editing. Functional studies in mutant iPSC-CMs showed aberrant calcium homeostasis, impaired contractility, and sarcomeric disorganization, recapitulating the hallmarks of DCM. By combining genetic analysis with human iPSCs, genome editing, and functional assays, we identified *SPEG* E1680K as a novel mutation associated with early onset DCM and provide evidence for its pathogenicity *in vitro*. Our study provides a conceptual paradigm for establishing genotype-phenotype associations in DCM with autosomal recessive inheritance.

## Introduction

Dilated cardiomyopathy (DCM) is a complex disorder with a prevalence of 1 in 250–400 individuals and a leading cause of heart failure [1]. Characterized by left ventricular dilation and systolic dysfunction, DCM is the most common form of cardiomyopathy and cause of heart transplantation in both adults and children [2]. Familial DCM, now more commonly

**Funding:** The funders had no role in study design, data collection and analysis, decision to publish, or preparation of the manuscript.

**Competing interests:** The authors have declared that no competing interests exist.

diagnosed, accounts for up to half of reported cases [3]. In the past two decades, the realization of the significance of a familial link in DCM has led to the discovery of a plethora of mutations in genes encoding proteins involved in various functions of the myocardium. To date, at least 50 genes have been implicated in DCM causation or risk, including genes encoding cytoskeletal, sarcolemma, mitochondrial, calcium cycling, costamere, and sarcomere proteins [1,4].

While these discoveries have provided invaluable insights, our understanding of the genetic basis of familial DCM remains incomplete. First, the genetics of DCM are characterized by locus and allelic heterogeneity, with most mutations being rare or even unique ('private') [5]. Given the low frequency and locus and allelic heterogeneity, attempts to validate the pathogenicity of putative disease-causing gene variants in a small number of probands or families have proven challenging. Second, the hitherto reported mutations explain only a fraction of familial DCM [6], primarily those exhibiting autosomal dominant inheritance [7]. Meanwhile, autosomal recessive inheritance has been implicated in pediatric DCM [7–9]. Finally, the growing number of gene variants linked to DCM, combined with insufficient evidence of the pathogenicity of each mutation presents a dilemma at the bedside, complicating the diagnosis and treatment of symptomatic patients and the management of asymptomatic relatives [7,10,11].

The recent advent of innovative technologies, such as human induced pluripotent stem cells (iPSCs) [12] and CRISPR/Cas9-based genome editing [13], and an increasingly refined capacity to differentiate iPSCs into cardiomyocytes (iPSC-CMs) [14] provide an opportunity for the generation of human patient-specific cells for disease modeling. Human iPSC-CMs exhibit fetal-like phenotypes, which are often seen as a drawback for modeling late onset diseases [15]. Despite this immaturity, iPSC-CMs represent a suitable model for studying monogenic diseases that manifest phenotypes during fetal or early postnatal stages of development, such as those observed in the family in our study.

In this study, we report a novel autosomal recessive variant (p. E1680K) in striated muscle enriched protein kinase (*SPEG*) causing early onset DCM. Combining iPSC and CRISPR/Cas9-mediated genome engineering technologies, we established an *in vitro* model to ascertain the pathogenicity of this mutation. Functional analysis showed that iPSC-CMs carrying the homozygous *SPEG* E1680K mutation recapitulated the hallmarks of DCM, including aberrant calcium homeostasis, impaired contractility, and sarcomeric disarray. These observations suggest *SPEG* E1680K as a novel DCM-causing mutation. Our study is the first to report *SPEG* E1680K as a human genetic variant and to validate its pathogenicity in recessive, early-onset DCM. We provide a proof-of-concept that *in vitro* iPSC models could effectively validate the pathogenicity of novel genetic variants associated with recessively inherited DCM that are otherwise poorly supported by few probands and families.

## Materials and methods

### Derivation of human induced pluripotent stem cells

All protocols for this study were approved by the Stanford University Institutional Review Board. Peripheral blood mononuclear cells (PBMCs) were obtained by conducting a standard blood draw from consenting individuals and isolated using a Ficoll gradient separation. PBMCs were reprogrammed using a Sendai virus vector expressing OCT4, KLF4, SOX2, and MYC (OKSM) (Life Technologies) following the protocol supplied by the manufacturer. Approximately one month after reprogramming, iPSC clones were isolated and cultivated on growth factor-reduced Matrigel (Corning)-coated 6-well tissue culture dishes (Greiner) in E8 pluripotent stem cell culture medium (Life Technologies).

## Chemically-defined differentiation of human iPSC-CMs

The iPSCs were differentiated into iPSC-CMs using a small molecule-mediated protocol [16]. Briefly, iPSCs were cultured in RPMI 1640 medium supplemented with 1x B27 minus insulin supplement (Thermo Fisher Scientific) and the small molecule CHIR99021 (6 μM) for 48 hours. The media was then switched to RPMI 1640 medium supplemented with 1x B27 minus insulin supplement and the small molecule IWR-1 (3 μM) for another 48h. Twelve days after cardiac differentiation, iPSC-CMs were enriched with RPMI-1640 without glucose (Life Technologies) supplemented with 1x B27 minus insulin and 5 mM sodium DL-lactate (Sigma) for 96h. Beating iPSC-CMs were maintained in RPMI 1640 medium supplemented with 1x B27 supplement (Thermo Fisher Scientific). Dissociation of iPSC-CMs was performed using pre-warmed TrypLE select 10x (Thermo Fisher Scientific) at 37˚C for 10min. After detaching, cells were collected by centrifugation (100g, 5 min), resuspended in RPMI 1640 with 1x B27 media, and plated in Matrigel-coated dishes. All experiments were performed with iPSC-CMs at forty-five to sixty days post differentiation.

## CRISPR/Cas9-mediated genome editing

A guide RNA targeting the *SPEG* locus was designed using a web-based tool (crispr.mit.edu/) and chosen based on a high on-target score and the lowest off-target score. The guide sequence was cloned into the single guide RNA (sgRNA) scaffold of the BsbI-digested pSpCas9(BB)-2A-GFP vector (PX458, a gift from Feng Zhang; Addgene plasmid #48138; http://n2t.net/addgene:48138; RRID:Addgene_48138) using a pair of annealed oligos. Genome editing was performed by co-transfection of iPSCs with the CRISPR/Cas9/gRNA vector (1 μg) and homology-directed repair (HDR) single-stranded oligo DNA nucleotides (ssODN) (4 μg) using the Lipofectamine Stem Reagent (Thermo Fisher Scientific). Twenty-four hours after transfection, cells were dissociated into single cells using TrypLE Express (Thermo Fisher Scientific) and sorted by flow cytometry for GFP expression. The GFP positive cells were seeded at a density of 1,000 cells per well in a Matrigel-coated 6-well plate for single cell clone expansion. Seven to ten days after seeding, individual clones were manually picked into a 96-well plate. A small proportion of the cells were used for DNA extraction with QuickExtract solution (Epicenter), followed by PCR using the PrimeSTAR GXL DNA Polymerase (Takara) and site-specific primers (Forward: cccatagaaagtgggagttcaag; Reverse: actctgctgcctacctaaca). Primers and oligos were synthesized at Integrated DNA Technologies (IDT). We isolated isogenic clones from the targeted parental iPSC line: an unedited control (referred to as SPEG$^{WT}$) clone and a genome-edited homozygous (referred to as SPEG$^{MUT}$) clone.

gRNA: tctgggatgtagctgcacagagg;

HDRssODN:gtgctgagctgggacctgccctgagcgctgggctgggccgggcagttggcactgggcactgttctccttgatctgggatgtagctgcacagaggagctgctggagcgaatcgccaggaaacccaccg.

## Off-target detection

Genomic DNA was extracted from the parental and the gene-edited iPSCs using the DNeasy Blood & Tissue Kit (Qiagen). Primers were designed to amplify regions corresponding to the top ranking *in silico* predicted off-target sites using the bioinformatics tool COSMID. The top 10 targets were selected, and putative off-target sites were amplified by PCR using 1.25 units of PrimeSTAR GXL DNA Polymerase (Clontech), 50 ng of genomic DNA, and 0.4 μM of forward primer and 0.4μM of reverse primer in a total volume of 25 μl per reaction. The PCR amplicons were sequenced by the Sanger method. Primer sequences are shown in S1 Table.

## Immunocytochemistry

For pluripotency marker analysis, human iPSC colonies grown in Matrigel-coated 8-well chamber glasses (Thermo Scientific) were fixed using 4% paraformaldehyde (PFA) for 10 min at room temperature. Cells were permeabilized with 0.5% Triton X-100 followed by blocking with 5% goat serum in PBS with 0.1% Tween 20. The cells were incubated with mouse anti-SSEA4 (R&D systems), rabbit anti-OCT3/4 (Santa Cruz Biotechnology), or mouse anti-SOX2 (R&D systems) primary antibodies overnight, washed and then incubated with Alexa Fluor-conjugated secondary antibodies (Life Technologies) for 1 h at room temperature. Finally, the cells were counterstained with DAPI (Life Technologies) and mounted with anti-fade medium (DAKO). For sarcomere protein visualization, primary antibodies include rabbit polyclonal anti-cardiac troponin T (Abcam) and mouse monoclonal anti-sarcomeric alpha-actinin (Sigma-Aldrich). Epifluorescence Images were acquired using an Eclipse 80i microscope (Nikon Instruments). High resolution images were acquired using a Zeiss LSM 880 confocal system with airyscan imaging mode, followed by airyscan processing using Zeiss Zen software (Zeiss).

## Sarcomere length and sarcomere packing density

The length of the sarcomeres and their spatial organization were calculated based on the 'sarcomere packing density' concept [17], which was implemented in a custom MATLAB script (Mathworks). Briefly, we defined a square region of 200x200 pixels in each image of well-defined iPSC-CMs immunostained for alpha-actinin and calculated the 2D Fourier power spectrum. Next, we selected a 30-degree angular portion of the 2D spectrum centered along the preferential orientation of the sarcomeres. The selected points are used to obtain the 1D profile of the power spectrum. In the presence of well-organized sarcomeres, this 1D profile exhibits an exponentially decaying signal with periodic peaks centered at multiples of the spatial frequency of the sarcomeres ($f_0$). The 1D power spectrum can be approximated as:

$$\Gamma(f, \zeta) = \Gamma_{ap}(f, \zeta) + \Gamma_p(f, \zeta) \quad \zeta = \{a_i, b_i, f_0\}_{i=0,1,2}$$

$$\Gamma_{ap}(f, \zeta) = a_0 \exp\left(-\frac{f}{b_0}\right)$$

$$\Gamma_p(f, \zeta) = \sum_{k=1}^{2} a_k \exp\left(-\frac{(f - k f_0)^2}{b_k}\right)$$

Where $\Gamma_{ap}(f, \zeta)$ represents an aperiodic noise term, i.e. the amount of non-periodic structure, and $\Gamma_p(f, \zeta)$ models the peaks related to periodically repeating structures, i.e. the sarcomeres. The parameters of the model ($\zeta$) were fitted with MATLAB's "lsqnonlin" function. The fitted parameters can be used to calculate the Sarcomeric Length (SL), and the Sarcomere Packing Density (SPD):

$$SL = f_0^{-1}$$

$$SPD = \frac{\int_0^\infty \Gamma_p(f, \zeta) df}{\int_0^\infty \Gamma(f, \zeta) df}$$

The custom MATLAB script is provided in the Supplementary Information (Supplementary note).

## Whole exome sequencing

Genomic DNA was extracted from peripheral blood and submitted to Otogenetics Corporation (Norcross, GA USA) for exome capture and sequencing. Briefly, genomic DNA was subjected to agarose gel and OD ratio tests to confirm the purity and concentration prior to Covaris (Covaris, Inc., Woburn, MA USA) fragmentation. Fragmented genomic DNAs were tested for size distribution and concentration using an Agilent Tapestation and Nanodrop. Illumina libraries were made from qualified fragmented genomic DNA using NEBNext reagents (New England Biolabs), and the resulting libraries were subjected to exome enrichment using NimbleGen SeqCap EZ Human Exome Library v2.0 (Roche NimbleGen) following manufacturer's instructions. Enriched libraries were tested for enrichment by qPCR and for size distribution and concentration by an Agilent Bioanalyzer 2100. The samples were then sequenced on an Illumina HiSeq2000 which generated paired-end reads of 90 or 100 nucleotides (nt). Data was analyzed for data quality, exome coverage, and exome-wide SNP/InDel using the platform provided by DNAnexus (DNAnexus, Inc).

## SNP karyotyping

SNP karyotype analysis was performed on the Illumina's CytoSNP-850K genotyping microarrays, which measure approximately 850,000 SNPs across the genome. All genomic DNA was isolated from iPSC clones according to the manufacturer's protocol (Qiagen). Input genomic DNA (500 ng) was processed, hybridized to the array and scanned on an Illumina HiScan according to the manufacturer's instructions. Copy number variations (CNVs) were identified using the cnvPartition Pluginv.3.2.0 in GenomeStudio (Illumina) by assessing both the B-allele-frequency and Log R ratios.

## Quantitative RT-PCR

Total RNA was isolated from iPSC-CMs using the Qiagen RNeasy Mini kit. 1 µg of RNA was used to synthesize cDNA using the iScript cDNA Synthesis kit (Bio-Rad). 0.25 µl of the reaction was used to quantify gene expression by qPCR using TaqMan Universal PCR Master Mix. Expression values were normalized to the average expression of housekeeping gene 18s.

## Intracellular Ca$^{2+}$ imaging and analysis

For ratiometric calcium imaging, dissociated iPSC-CMs were seeded on Matrigel-coated coverslips (CS-24/50, thickness 1 mm, Warner Instruments). After 7 days, cells were loaded with 5 µM Fura-2AM (Thermo Fisher Scientific) with 0.02% Pluronic F-127 (Thermo Fisher Scientific) in Tyrode's solution for 10 min at room temperature, followed by washing with Tyrode's solution. iPSC-CMs were field-stimulated at 0.5 Hz at 37°C. Single cell Ca$^{2+}$ imaging was conducted on a Nikon Eclipse Ti-E inverted microscope with a 40x oil immersion objective (0.95 NA). Fura-2 was excited at 340 nm and 380 nm wavelengths using a Lambda DG-4 ultra-high-speed wavelength switcher (Sutter Instrument), and the emission was collected at 510 nm wavelength.

Raw data were analyzed using a custom Python script (https://github.com/GeorgeMcMullen/CalciPy) built specifically to automate processing of ratiometric calcium imaging data [16].

## Contractility assays

Throughout the study, contraction analysis was performed using a high-speed video microscopy with motion vector analysis to investigate the contractile characteristics of iPSC-CM monolayers. Depending on the design of the experiment, iPSC-CMs were seeded at $2x10^5$ cells/cm$^2$ into 384-well or 96-well plates. Contractility was measured using the Sony SI8000 cell motion imaging system. Video imaging of beating iPSC-CMs was recorded for 10 sec at a frame rate of 75 fps, a resolution of $1024 \times 24$ pixels, and a depth of 8 bits using a $10\times$ objective on a fully automated Nikon microscope (Eclipse Ti, Nikon). Motion detection and analysis were performed using the Sony Cardio-analysis software (based on a block matching algorithm) [18].

## Preparation and culture of engineered heart tissues (EHTs)

EHTs were prepared using a fibrin gel as previously described [19]. Briefly, EHTs were produced around silicone posts in agarose casting molds in a 24-well plate. Each EHT contained $1x10^6$ iPSC-CMs in a fibrin matrix (100 μL total) composed of 10 μL Matrigel (Corning), 5 mg/ml bovine fibrinogen (2.53 μL of 200 mg/ml fibrinogen reconstituted in 0.9% NaCl) and supplemented with 0.1 mg/mL aprotinin (Sigma Aldrich) and 3 U/ml thrombin (Sigma Aldrich). EHTs were incubated for 2 hours at 37°C and then transferred to a new plate containing RPMI supplemented with B27 and 10% KnockOut serum replacement (Thermo Fischer Scientific). The medium was replaced every two days with RPMI supplemented with B27 and 33 μg/ml aprotinin.

## Contractile analysis of EHTs

EHT contractile motion was recorded using the SI8000 Cell Motion Imaging System (Sony). The SI8000R Analyzer Software (Sony) was used to detect motion vectors and track the movement of each EHT post. Maximum post deflection from rest was extracted from each contraction cycle in the tracking data using a custom python script. Calculation of contractile force is a function of average distance travelled from each resting phase to its peak of contraction, combined with the physical properties of the posts considering an elastic modulus of 1.7 MPa, a post radius of 0.5 mm and a distance between posts (length) of 10 mm based on the equation:

$$contractile\ force\ =\ \frac{3\pi\ [distance\ travelled][elasticity\ modulus][post\ radius]^4}{4[length]^3}$$

## Statistical analysis

Comparison between iPSC-CMs carrying the E1680K mutation and the isogenic controls was performed using unpaired Student's t-test. The criterion for significance was set at $p < 0.05$.

# Results

## Clinical presentation of early onset DCM

We have identified a family with consanguineous marriage from the Israeli Bedouin population of the Negev, who are affected by early-onset, severe DCM in an autosomal recessive pattern of inheritance (Fig 1). Five family members presented with acute DCM with clinical onset varying from prenatal (33 weeks of gestation) to five years of age. All five patients have died

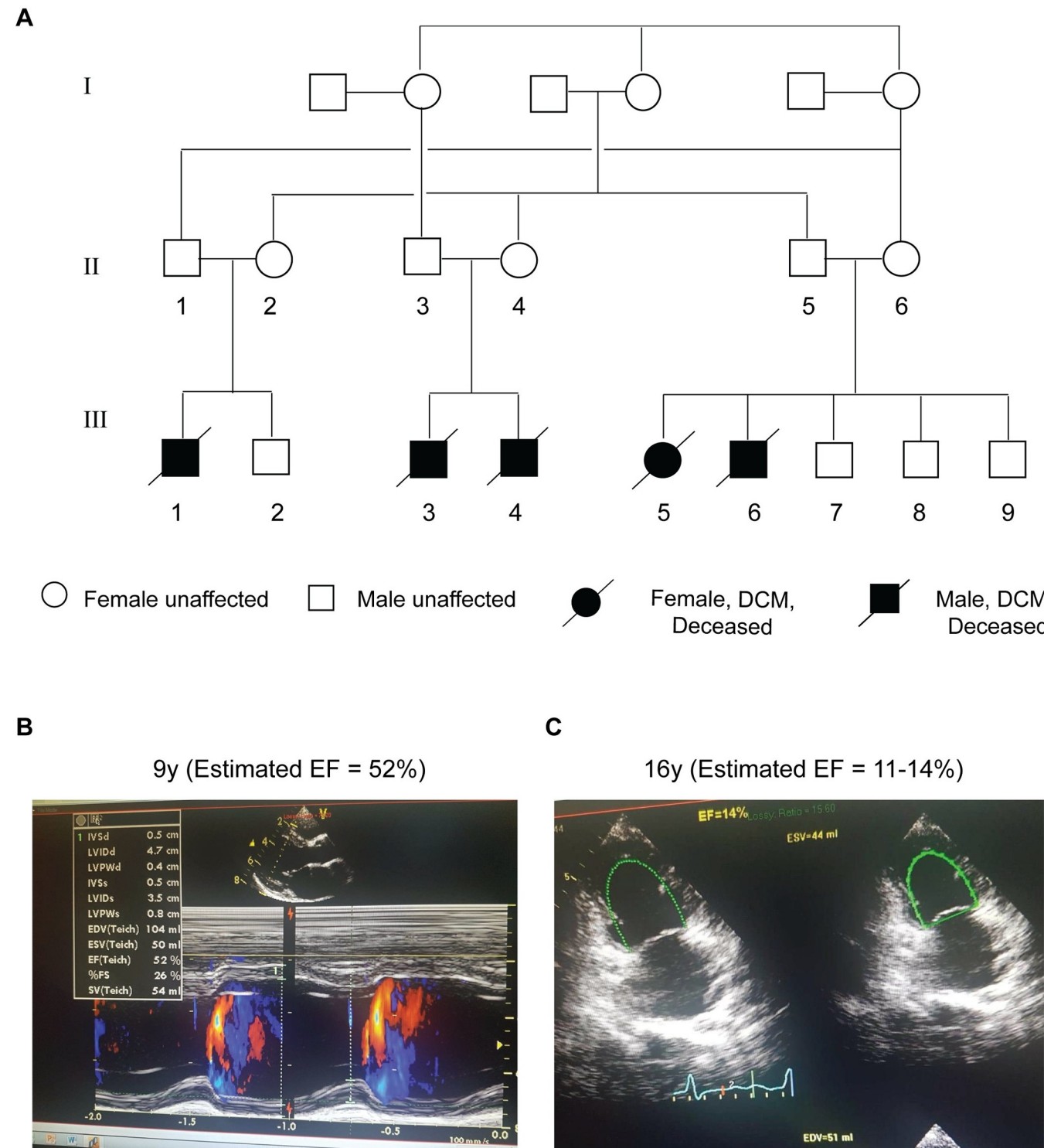

**Fig 1. Family pedigree and DCM imaging.** (A) Schematic pedigree of the consanguineous family affected by early-onset, severe DCM in an autosomal recessive pattern of inheritance. Circles represent female family members, and squares represent males. Filled symbols indicate family members diagnosed with DCM, diagonal lines indicate death. All the family members with open symbols underwent complete physical and cardiac evaluation and were found to be healthy. (B-C) Echocardiographic images of Patient III-5 showing the DCM progression. Left: an exam obtained in 2012 at the age of 9 years. M-mode tracing of the LV dimensions over time obtained in the parasternal long axis view. Estimated EF was 52% indicating mild LV dysfunction. Right: an exam obtained in 2019 at the age of 16 years few days before death. Systolic and diastolic LV dimensions are shown in apical 2 chamber view. The estimated EF in this exam was 14% indicating extremely severe LV dysfunction.

due to heart failure. The clinical presentation of each patient is detailed below and summarized in Table 1.

Patient III-1: First male child of healthy, consanguineous parents of Bedouin origin. Diagnosed with DCM at 33 weeks of gestation. Primary clinical features included respiratory distress, congestive heart failure (CHF), cardiogenic shock, and supraventricular tachycardia. Echocardiogram revealed severe LV dilation, reduced LV fractional shortening (LVFS), and mitral regurgitation (MR). Patient died at the age of two months.

Patient III-3: First male child of healthy, consanguineous parents of Bedouin origin, first cousin of Patient III-1. Documented to have normal cardiac size prior to DCM onset. Diagnosed with DCM at the age of five years after presentation to clinic with respiratory distress, followed by frequent hospitalization due to CHF. Patient exhibited normal cognitive development. Echocardiogram revealed severe LV dilation, reduced LVFS, moderate left ventricular hypertrophy (LVH), MR, and pulmonary hypertension (PH). Patient died at the age of 12 years.

Patient III-4: Younger brother of Patient III-3. Diagnosed with DCM at the age of two years after presentation to clinic with respiratory distress, followed by frequent hospitalization due to CHF. Patient achieved normal developmental milestones. Echocardiogram revealed severe LV dilation, reduced LVFS, severe MR, moderate LVH, moderate PH, and small muscular ventricular septal defect (VSD). Patient died at the age of four years.

Patient III-5: First female child of healthy, consanguineous parents of Bedouin origin, first cousin of Patient III-1 and III-3. Documented to have normal cardiac size and LVFS prior to DCM onset. Diagnosed with DCM at the age of one after presentation to clinic with respiratory distress and CHF during first year of life. Echocardiogram revealed moderate LV dilation, mildly reduced LVFS, moderate LVH, and moderate secundum atrial septal defect (ASD). Patient achieved normal developmental milestones and remained stable until 12 years of age, although follow-up revealed progressive deterioration of cardiac function (Table 1 and Fig 1B and 1C). Patient was frequently hospitalized due to CHF exacerbation and eventually died at the age of 16 years.

**Table 1. Clinical data of the DCM siblings.** The annotation of the patients is according to Fig 1. Abbreviations: CHF, congestive heart failure; FS, fractional shortening; LV, left ventricle; LVEDD, left ventricle end-diastolic diameter; LVH, left ventricle hypertrophy; MVI, mitral valve insufficiency; f/u, follow up; VSD, ventricular septal defect; MR, mitral regurgitation; ASD, atrial septal defect; SVT, supraventricular tachycardia. CK, serum Creatinine Kinase level (U/L—normal local laboratory range 20–180).

| Case | Age at onset | Age at death | Sex | Primary clinical features and follow-up (f/u) | Treatment | Echocardiographic interpretation | LVEDD (mm) | LV FS (%) |
|---|---|---|---|---|---|---|---|---|
| III 1 | 33W of gestation | 2M | M | Respiratory distress, CHF, cardiogenic shock, arrhythmia (SVT). | Digoxin, Thambucor, Lasix | Severe LV dilatation, severe LV dysfunction, severe MR | 36 | <10 |
| III 3 | 5Y | 12Y | M | Respiratory distress. f/u–frequently hospitalized due to CHF, severe peripheral edema. Refusal to heart transplantation. CK—38 | Digoxin, Carvedilol, Captopril, Aldactone, Lasix | Severe LV dilatation, severe LV dysfunction, moderate LVH, moderate MR, pulmonary hypertension | 45 (5Y), 68 (11Y) | 23 (5Y), 20 (11Y) |
| III 4 | 2Y | 4Y | M | Respiratory distress. f/u–Frequently hospitalized due to CHF. CK—49 | Digoxin, Captopril, Lasix | Severe LV dilatation, moderate to severe LV dysfunction, severe MR, moderate LVH, moderate pulmonary hypertension, small muscular VSD | 48 (2Y) | 12–15 (2Y) |
| III 5 | 1Y | 16Y | F | Respiratory distress, CHF during first year of life. f/u- in a stable condition until 12Y of age. Later, frequently hospitalized due to CHF. Refusal to heart transplantation. CK—30 | Digoxin, Carvedilol, Lasix | Moderate LV dilatation, mild LV dysfunction, moderate LVH, moderate ASD II. f/u–severe LV dilatation, severe LV dysfunction | 23 (1M), 47 (9Y), 52 (11Y), 58 (16Y) | 30 (1M), 26 (9Y), 17 (11Y), 11 (16Y) |
| III 6 | 2Y | 5Y | M | Respiratory distress. f/u—Frequently hospitalized due to CHF. Refusal to heart transplantation. CK—28 | Digoxin, Captopril, Lasix | Severe LV dilatation, severe LV dysfunction, moderate LVH, moderate MVI, pulmonary hypertension | 50 (2Y) | 15–17 (2Y) |

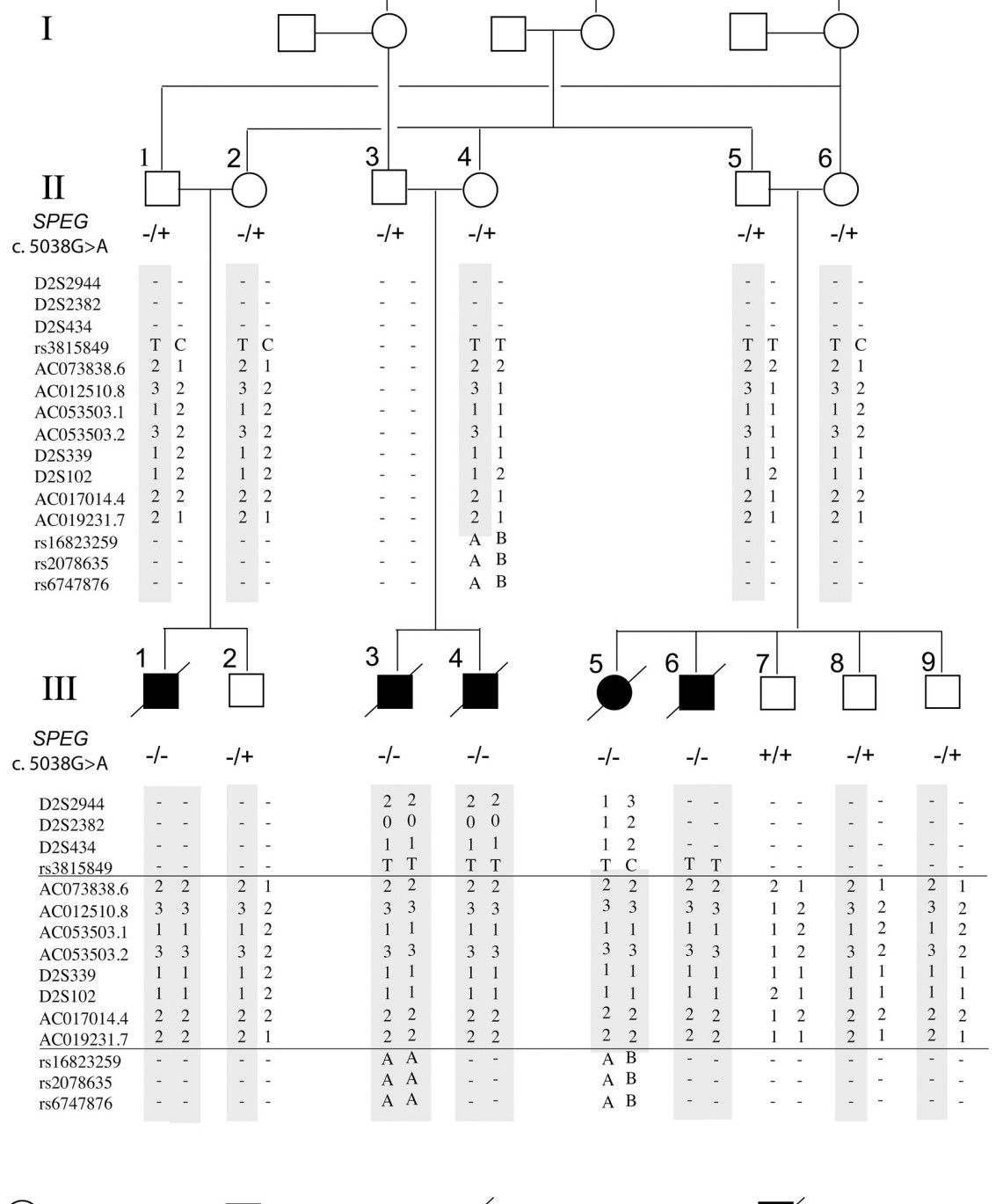

**Fig 2. Haplotype analysis in the consanguineous family.** The haplotype analysis based on microsatellite markers from 2q35-q36 revealed a founder haplotype (grey bar) for which the patients are homozygous. The horizontal lines mark the linkage border. All the family members whose haplotypes appear as open symbols underwent complete physical and cardiac evaluation and were found to be healthy. SPEG c. 5038G>A: -/- homozygous for the variation, -/+ heterozygous, +/+ reference allele.

Patient III-6: Younger brother of Patient III-5. Documented to have normal cardiac size prior to DCM onset. Diagnosed with DCM at the age of two years after presentation to clinic with respiratory distress, followed by frequent hospitalization due to CHF. Patient achieved normal developmental milestones. Echocardiogram revealed severe LV dilation, reduced LVFS, moderate LVH, moderate MR, and PH. Patient died at the age of five years.

Two of the patients, III-4 and III-5, had associated congenital heart defects, VSD and ASD, respectively. Given that VSDs and ASDs are relatively common types of heart defects that present at birth [20] and were not seen in all affected patients, it is unlikely that these congenital defects are associated with the SPEG mutation. However, we cannot totally rule out this possibility. Notably, the normal cardiac size and function documented for two of the affected children prior to the clinical onset of DCM (III-3 and III-6) supports the notion that myocardial developmental defects unlikely contributed to the pathogenesis of DCM. Finally, the serum creatine kinase levels were evaluated in four of the five affected children and were in the normal range (Table 1), consistent with the clinical evaluation of normal skeletal muscle function in all the affected patients.

## Linkage analysis and homozygosity mapping

The recessive pattern of DCM inheritance in the consanguineous family suggests homozygosity of a pathogenic mutation inherited from a common founder. In order to identify the disease-causing mutation, we first combined linkage analysis and homozygosity mapping to locate the genomic region, or linkage interval, that is associated with the disease. The linkage interval to be identified should show homozygosity in all DCM patients, and heterozygosity or homozygosity for a different allele in unaffected family members.

Patients III-3, III-5, and the mother of patient III-3 (II-4, healthy) were genotyped with the Affymetrix GeneChip Human Mapping 250K Sty Arrays and analyzed using the KinSNP software. Four chromosomal segments larger than 4 centiMorgan (cM) were shared by the two patients analyzed. Variable number tandem repeat (VNTR) analysis revealed heterozygosity in three of these segments in Patients III-3 and III-5, negating their involvement in pathogenesis. In contrast, all five affected patients in the family show homozygosity for eight different VNTR markers in the fourth segment (Fig 2). This 9.4cM segment is located at chr2: 218,317,008–227,728,735 (Assembly: Mar. 2006 NCBI36/hg18) and delimited by VNTR markers AC073838.6 and AC019231.7. Primers used for PCR amplification of the VNTRs are listed in S2 Table. Statistical analysis using the PedTool Server yielded a two-point Lod Score of 3.66 for the fully informative AC053503-2 marker (Table 2) and a multiple-point Lod Score of 5.25, confirming linkage of this interval.

## Exome sequence analysis

In order to identify mutation(s) that may have contributed to the development of DCM in the affected family members, we conducted whole exome sequencing in patient III-6 and

**Table 2. Two-point analysis Lod-Score.** The Lod-Score was calculated using the Pedtool server, assuming recessive inheritance with 99% penetrance and an incidence of 0.01 or 0.001 for the disease allele.

| Marker information | | Recombination fraction | | | | | | |
|---|---|---|---|---|---|---|---|---|
| Marker id | Marker name | 0.00 | 0.01 | 0.05 | 0.10 | 0.20 | 0.30 | 0.40 |
| 1 | AC053503-1 | 2.1686 | 2.1152 | 1.9027 | 1.6409 | 1.1397 | 0.6832 | 0.2926 |
| 2 | AC053503-2 | 3.6684 | 3.5887 | 3.2679 | 2.8644 | 2.0586 | 1.2750 | 0.5490 |
| 3 | D2S339 | 0.6507 | 0.6295 | 0.5465 | 0.4476 | 0.2726 | 0.1377 | 0.0486 |
| 4 | D2S102 | 2.3420 | 2.2877 | 2.0709 | 1.8017 | 1.2773 | 0.7819 | 0.3297 |
| 5 | AC017014-4 | 1.6271 | 1.5883 | 1.4332 | 1.2400 | 0.8621 | 0.5070 | 0.1998 |

identified four recessive variants within the linkage interval on chromosome 2: *C2orf24*, *DNPEP*, *DOCK10*, and *SPEG* (Table 3). To further explore whether these mutations are disease-causing or disease-associated, we examined their prevalence in the general population, conservation across species, and functional consequences by *in silico* predictions.

The *C2orf24* (rs4674361) and *DNPEP* (rs907679) variants are single nucleotide polymorphisms (SNPs) with high allele frequency in the general population and are therefore unlikely to be pathogenic. A third recessive variant in *DOCK10* (rs147752392) results in a glycine to arginine missense mutation (p.G570R). This variant is reported in the Genome Aggregation Database (GnomAD v2.1), the NHLBI Trans-Omics for Precision Medicine (TOPMed) Whole Genome Sequencing Program dataset, and the ExAC database with allelic frequency of 0.005122, 0.002915, and 0.00728, respectively. Notably, the GnomAD database includes nine homozygous *DOCK10* p.G570R healthy carriers, negating the pathogenicity of this variant.

Finally, the fourth variant found in exon 23 of the *SPEG* gene (c. 5038G>A, p. E1680K, NM_005876) has not been reported in the GnomAD or TOPMed databases and therefore is unlikely to be a common polymorphism. All family members were Sanger sequenced for this variant which segregated as expected in the family (Fig 2). According to bioinformatic prediction, the *SPEG* gene has a probability of being loss of function intolerant (pLI) score of 1.00 and is predicted to be highly intolerant to variation [21]. The *SPEG* variant identified in our index family results in a missense mutation (p.E1680K) in the first of two SPEG serine/threonine protein kinase domains (Fig 3A and 3B). The amino acid residue implicated in this variant is highly conserved even across species (Fig 3C), further suggesting that changes in this residue are likely to be detrimental. Taken together, we concluded that *SPEG* E1680K is likely the pathogenic variant in our index family.

## Generation of *SPEG* E1680K iPSC-CMs

We sought to validate the pathogenicity of *SPEG* E1680K in human iPSC-CMs. Using a CRISPR/Cas9-based genome editing approach we introduced the homozygous *SPEG* E1680K mutation in an iPSC line (hereafter referred to as SPEG^MUT) derived from a healthy individual (Fig 4A and 4B). Another clone that was unmodified at the SPEG locus was isolated from the CRISPR-targeted parental line as a wildtype control (hereafter referred to as SPEG^WT). These genetically engineered iPSCs lines differ exclusively at the edited locus and were otherwise isogenic. The genome-edited iPSCs were karyotypically normal and maintained their pluripotency as assessed by immunofluorescence staining for pluripotency markers (S1 Fig). Importantly, the CRISPR/Cas9-mediated genome editing approach did not introduce any unwanted alteration in the genome as assessed by genotyping of the top ten *in silico* predicted off-target loci (S2 Fig).

To assess the potential role of *SPEG* E1680K in cardiomyocyte physiology, we differentiated the SPEG^MUT and isogenic control SPEG^WT iPSCs into cardiomyocytes using established protocols that generate a nearly pure population (>90%) of cardiomyocytes. SPEG is a serine/threonine kinase that is highly expressed in the developing heart [22,23]. In agreement, we

**Table 3. Exome sequencing analysis.** Information of four variants from exome sequencing. The chromosome number and location, gene name, changes of the nucleotides and the amino acids, and frequencies from dbSNP and ExAc databases are shown.

| chr | location | Ref seq | Var seq | Gene name | Transcript name | Ref peptide | Var peptide | dbSNP | dbSNP Frequency | ExAc Frequency |
|-----|----------|---------|---------|-----------|-----------------|-------------|-------------|-------|-----------------|----------------|
| chr2 | 220037393–220037394 | a | g | C2orf24 | NM_015680 | L | P | rs4674361 | 0.0048 | 0.99 |
| chr2 | 220250147–220250148 | a | g | DNPEP | NM_012100 | V | A | rs907679 | 0 | 1 |
| chr2 | 220343876–220343877 | g | a | SPEG | NM_005876 | E | K | | | |
| chr2 | 225721677–225721678 | c | t | DOCK10 | NM_014689 | G | R | rs147752392 | 0.0018 | 0.0072 |

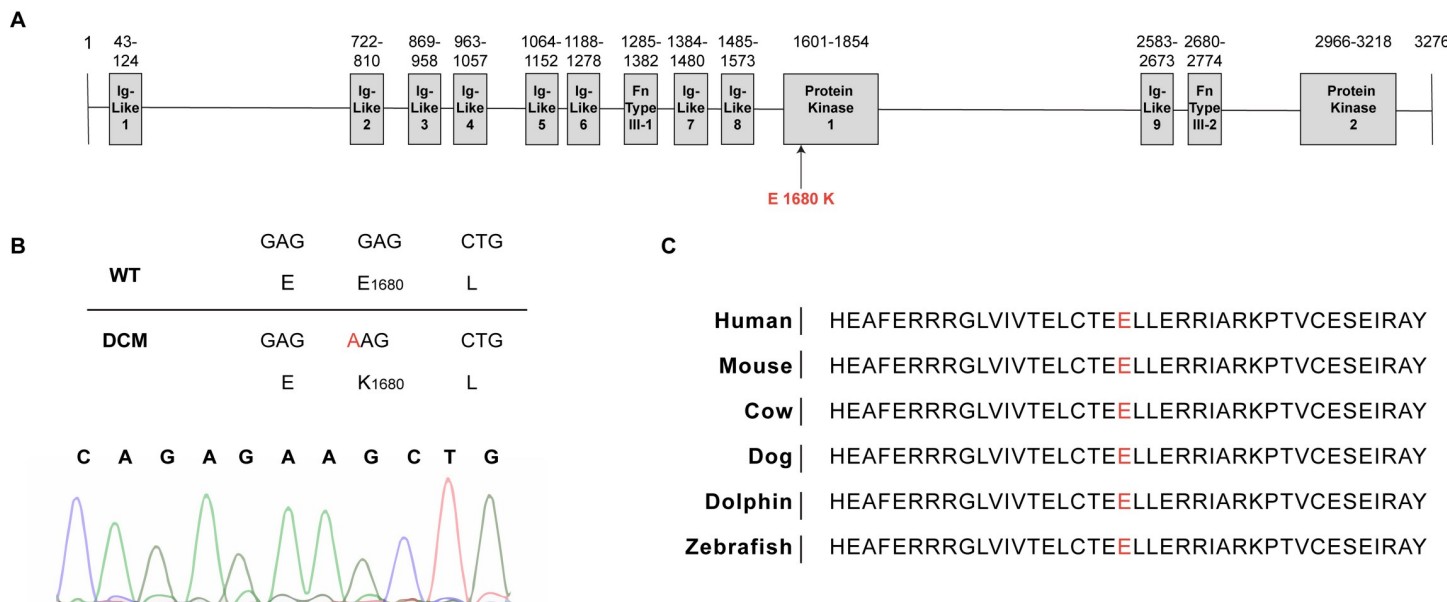

**Fig 3. SPEG domain organization and DCM associated mutation.** (A) Schematic overview of the SPEG protein structure showing the key functional domains and the position of a new SPEG variant identified in kinase domain 1. (B) Nucleotide sequencing of SPEG gene codon 1680 demonstrating homozygosity (G > A). (C) The Glutamic acid at position 1680 is highly conserved across species.

observed that the expression of SPEG was developmentally regulated with SPEG transcripts detected at an early stage of cardiac differentiation *in vitro*. Quantitative analysis showed comparable levels of SPEG mRNA expression between SPEG$^{MUT}$ and SPEG$^{WT}$ iPSC-CMs (S3 Fig). We attempted to quantify the expression of SPEG protein by Western blot analysis but it was unsuccessful because of lack of detection by commercially available antibodies. Next, we assessed the sarcomere integrity in the isogenic iPSC-CMs by immunofluorescence staining of cardiac troponin T and alpha-sarcomeric actinin to visualize the thin filaments and the z disk, respectively. Consistent with other iPSC-CM models of DCM [24–28], SPEG$^{MUT}$ iPSC-CMs exhibited disorganized myofibrils and abnormal, irregular sarcomeres with punctate staining, while SPEG$^{WT}$ iPSC-CMs showed well-aligned sarcomeres (Figs 4C and S4). We validated these observations by quantifying the 'sarcomere packing density' [17], a quantitative metric for the organization of the contractile cytoskeleton of iPSC-CMs (Figs 4D and S5).

Next, we investigated the effect of the E1680K mutation on cardiomyocyte function at the single-cell level. As aberrant calcium (Ca$^{2+}$) homeostasis is a hallmark of DCM, we examined the calcium handling properties of SPEG$^{MUT}$ and isogenic SPEG$^{WT}$ control iPSC-CMs. We observed significant increases in the Ca$^{2+}$-transient amplitude and faster Ca$^{2+}$ decay kinetics in SPEG$^{MUT}$ iPSC-CMs compared to the isogenic controls (Fig 5A and 5C), which were associated with a significant decrease in phospholamban gene (*PLN*) expression (Fig 5B), a critical regulator of Ca$^{2+}$ homeostasis in the heart. [29].

As DCM is characterized by a contractile defect, we next investigated the effect of the E1680K mutation on iPSC-CM contractility. Light microscopic video images of cardiomyocytes grown in monolayers were captured with a high-speed camera, and motion vectors were calculated with high spatiotemporal resolution using a block-matching algorithm [18]. Compared to SPEG$^{WT}$ isogenic controls, iPSC-CMs carrying the *SPEG* E1680K mutation exhibited a significant decrease in contractility (Fig 5D). We also examined the contractile properties of 3-dimensional (3D) EHTs, a micro-engineered model resembling human heart tissue that

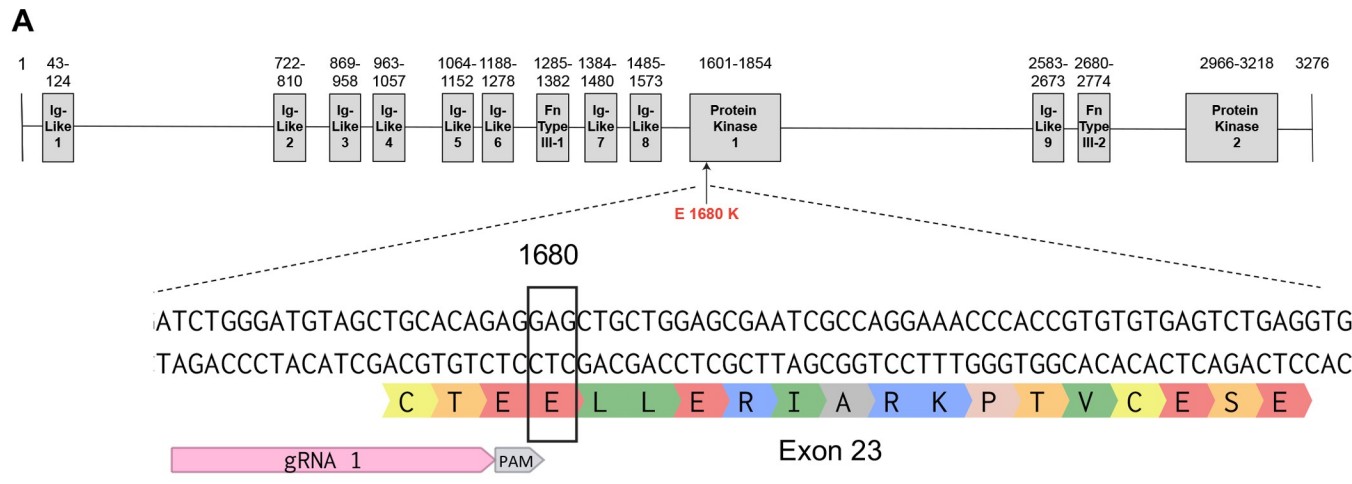

**Fig 4. CRISPR/Cas9-mediated engineering of iPSC-CMs carrying the SPEG E1680K mutation.** (A) Schematic of SPEG gene highlighting the target and Protospacer Adjacent Motif (PAM) for CRISPR/Cas9 editing. (B) Genotyping of human iPSCs showing the introduction of c. 5038G>A. (C) Immunostaining of cardiac alpha-actinin (green) and cardiac Troponin T (Red) in cardiomyocytes generated from isogenic control (WT) and SPEG mutant (E1680K) human iPSCs. Images are z projections. Scale bar = 10 μm. (D) Quantification of the sarcomere packing density in isogenic iPSC-CMs (n = 3 differentiation batches). Box-and-whisker plots show the minimum, the 25th percentile, the median, the 75th percentile, and the maximum. *P < 0.05.

exhibits a higher degree of maturation compared to monolayer cultures [19]. In agreement with the findings in the 2D monolayer cultures, SPEG$^{MUT}$ generated significantly less force than SPEG$^{WT}$ iPSC-CMs in 3D EHTs (Fig 5E). Taken together, our findings suggest that the genetically engineered *SPEG* E1680K human iPSC-CMs recapitulated the hallmark phenotypes associated with DCM, including sarcomeric disorganization, aberrant Ca$^{2+}$ homeostasis and contractile defects, suggesting that *SPEG* E1680K is a pathogenic variant.

## Discussion

In this study, we discovered and validated a novel autosomal recessive variant in the *SPEG* gene (p.E1680K) associated with early-onset DCM, implicating *SPEG* as a new candidate gene for DCM. By combining human iPSCs and CRISPR/Cas9-mediated genome editing technologies, we showed that iPSC-CMs carrying the *SPEG* E1680K mutation recapitulated DCM phenotypes *in vitro*. Our study demonstrated the potential of using genetically engineered human iPSC-CMs as a human cellular model to validate the pathogenicity of variants associated with autosomal recessive DCM.

*SPEG* encodes the striated muscle enriched protein kinase, a recently identified member of the junctional membrane complex of the muscle [30]. Mutations in *SPEG* have recently been identified in patients with centronuclear myopathies (CNMs), a group of disorders characterized by severe muscle weakness with respiratory impairment, ophthalmoplegia, and scoliosis. Some CNM patients carrying homozygous or compound heterozygous *SPEG* mutations have been diagnosed with DCM, corroborating our findings implicating the *SPEG* E1680K mutation in DCM [31,32]. Unlike CNM patients who present with severe myopathies, the affected individuals in our index family presented with fulminant DCM without any clinical manifestation of myopathy. These observations suggest that *SPEG* mutations may play a role in DCM pathogenesis in a non-syndromic manner independent of skeletal muscle involvement.

Consistent with our findings, disruption of the *Speg* gene causes a DCM-like phenotype and perinatal death in mice [23,30,33], supporting the notion that SPEG plays an important role in cardiac physiology and pathophysiology. Nevertheless, the role of SPEG in cardiac function is poorly understood. Recent studies suggest that SPEG regulates calcium re-uptake into the sarcoplasmic reticulum by phosphorylating Sarcoplasmic/Endoplasmic Reticulum Calcium ATPase 2 (SERCA2a) in cardiomyocytes [34]. Inducible deletion of *Speg* in the mouse heart was found to have a profound impact on cardiac function by inhibiting the calcium-transport activity of SERCA2a and impairing calcium re-uptake into the SR in cardiomyocytes. In agreement with a role of SPEG in calcium homeostasis we have observed a significant decrease in the mRNA expression levels of PLN in the *SPEG* E1680K mutant compared to control iPSC-CMs, supporting the hypothesis that *SPEG* E1680K alters calcium dynamics by decreasing the expression of PLN. Consistently, the calcium amplitude were significantly increased and the decay kinetics were significantly accelerated in *SPEG* mutant iPSC-CMs compared to isogenic controls. The phenotype is reminiscent of transgenic animal models of genetic DCM carrying mutations in the thin filament genes [35,36]. Notably, phosphorylation of SERCA2a by SPEG was previously reported as a function unique to the second kinase-domain of SPEG [34]. Therefore, how E1680K, a mutation in the first kinase-domain of SPEG, affects cardiomyocyte calcium homeostasis remains an interesting question for future investigation.

In summary, we discovered and validated a novel variant in the *SPEG* gene (p.E1680K) associated with nonsyndromic autosomal recessive, early-onset DCM. Importantly, we have demonstrated in this study the potential of combining human iPSC-CMs with genome editing technologies to translate clinical genetic findings into a biological understanding of the disease

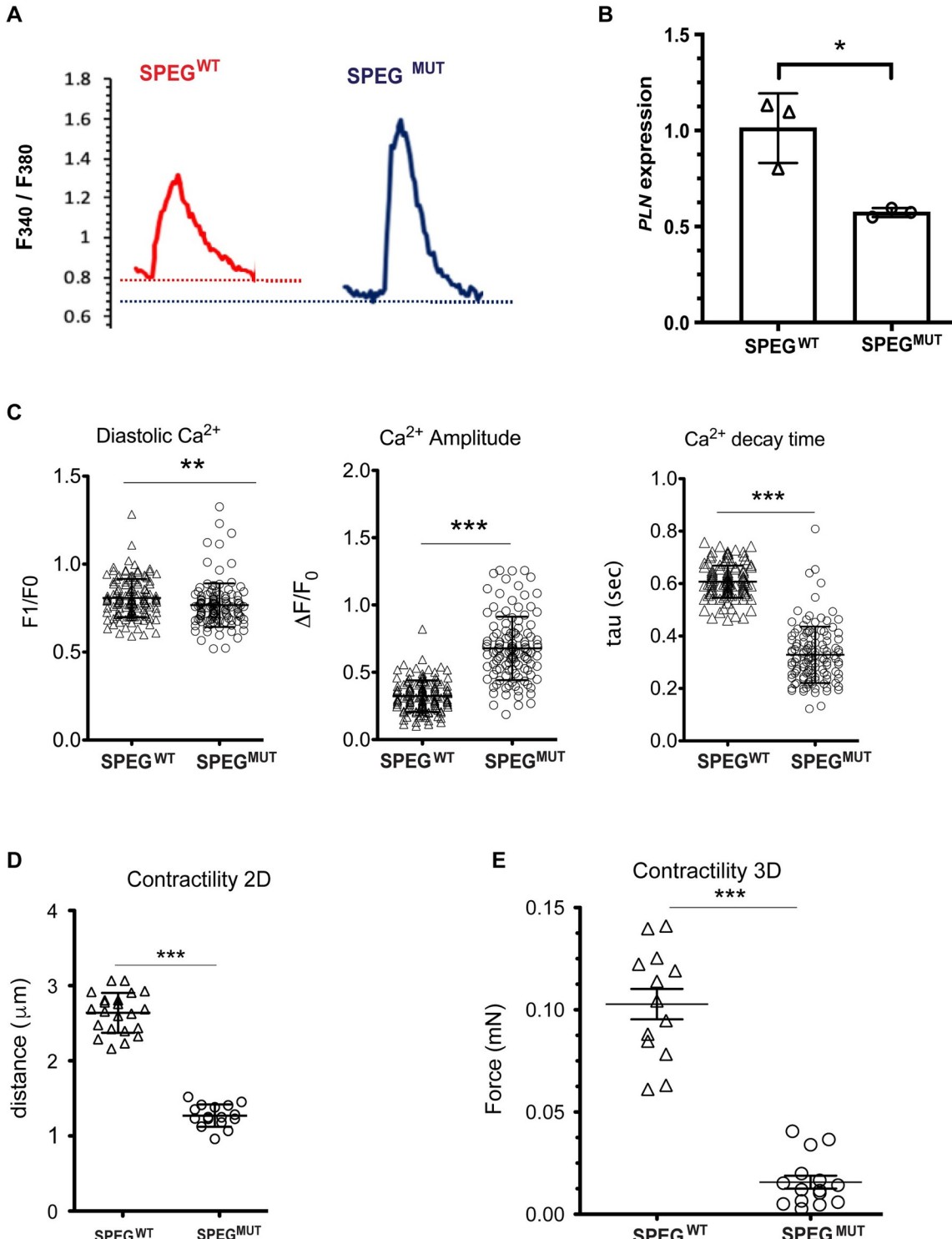

**Fig 5. Functional characterization of SPEG mutant iPSC-CMs.** (A) Representative intracellular calcium traces of isogenic control and SPEG E1680 mutant iPSC-CMs. (B) Expression of PLN mRNA in isogenic iPSC-CMs. Box-and-whisker plots show the minimum, the 25th percentile, the median, the 75th percentile, and the maximum. $^{*}P < 0.05$. n = 3 independent iPSC differentiation experiments. (C) Quantification of calcium kinetics parameters in isogenic control and mutant iPSC-CMs. Data represent mean ± SD; control n = 120 cells, SPEG mutant n = 130 cells from 3 independent iPSC differentiation experiments. $^{**}P < 0.01$, $^{***}P < 0.001$. (D) Contractility analysis in 2D monolayer preparation of isogenic control and SPEG E1680 mutant iPSC-CMs. Data represent mean ± SD; control n = 22, SPEG mutant n = 16 from 3 independent iPSC differentiation experiments. (E) Contractile force analysis of isogenic control and

SPEG E1680 mutant iPSC-CMs in 3D-EHTs. engineered heart tissues. Data represent mean ± SD; control n = 13, SPEG mutant n = 15 from 3 independent iPSC differentiation experiments. ***$P < 0.001$.

pathogenesis. This experimental framework combined with clinical segregation data could be broadly applied to distinguish genuine DCM disease-causing or disease-associated genetic variants from the broader genetic background where rare recessive variants are identified. A better understanding of the DCM etiology could ultimately improve our ability to care for patients at the bedside.

## Limitations

The major limitation of the current study is the lack of mechanistic links between the *SPEG* E1680K mutation and the pathogenesis of DCM, as well as the fact that homozygous variants are rare in DCM and *SPEG* heterozygous variants may not necessarily be disease-causing. Finally, the *in vitro* analyses of suspected pathogenic mutations using the iPSC-CM model can provide significant insights; however, these models do not always reflect the *in vivo* milieu.

## Ethics Statement

The study was approved by the Soroka Medical Center institutional review board. Human iPSCs were derived with approval from the Institutional Review Board at Stanford University (IRB-29904). All participants gave written informed consent prior to participation.

## Supporting information

**S1 Fig. Pluripotency of isogenic genome edited iPSCs.** (A) Representative immunofluorescence images of patient-specific iPSC colonies immunostained for the pluripotency-associated markers OCT-4, SOX-2, and SSEA-4. Scale bar = 100μm. (B) SNP-based karyotype analysis of the isogenic iPSCs showing no karyotypic abnormalities.
(TIF)

**S2 Fig. Analysis of off-targets events in isogenic CRISPR/Cas9 edited iPSCs.** The top ten in silico predicted sites were amplified by PCR and analysed by Sanger sequencing.
(TIF)

**S3 Fig. Relative SPEG mRNA expression levels.** The expression levels of SPEG were measured by qPCR analyses at different stages of cardiac differentiation of wild type (WT) and E1680K mutant (MUT) as indicated. The expression levels are normalized to TNNT2 expression. Mean ± SD, n = 3 independent differentiation experiments. (TIFF)
(TIF)

**S4 Fig. Assessment of sarcomere structure.** Representative immunofluorescent images of SPEG$^{WT}$ (A-C) and SPEG$^{MUT}$ (D-E) iPSC-CMs generated from 3 independent differentiation batches and used to quantify the sarcomere packing density associated with Fig 4D.
(TIF)

**S5 Fig. Quantification of Sarcomere Packing Density.** SPEG$^{WT}$ (panels A-D) and SPEG$^{MUT}$ (panels E-H). (A, E) Original image and selected region of interest (red square). (B, F) Enlarged image of the region of interest. (C, G) 2D Fourier power spectrum of the region of interest and selected orientation for 1D profile (blue wedges). (D, H) 1D profile of all data in the 2D spectrum (gray), selected orientation (blue), and fitted model (black).
(TIF)

**S1 Table. PCR Primers used for off target detection.**
(DOCX)

**S2 Table. Primers for PCR amplification of VNTRs and SNPs.**
(DOCX)

**S1 Note. Matlab Script to quantify sarcomere alignment.**
(M)

**S2 Note. Matlab GUI to quantify sarcomere alignment.**
(FIG)

## Acknowledgments

We are grateful to Prof. Val Sheffield for the genotyping of the family members.

## Author Contributions

**Conceptualization:** Michael S. Kapiloff, Yoram Etzion, Ruti Parvari, Ioannis Karakikes.

**Data curation:** Aviva Levitas, Emad Muhammad, Isaac Perea Gil, Yoram Etzion, Ruti Parvari, Ioannis Karakikes.

**Formal analysis:** Aviva Levitas, Emad Muhammad, Yuan Zhang, Isaac Perea Gil, Ricardo Serrano, Nashielli Diaz, Maram Arafat, Alexandra A. Gavidia, Yoram Etzion, Ruti Parvari, Ioannis Karakikes.

**Funding acquisition:** Yoram Etzion, Ruti Parvari, Ioannis Karakikes.

**Investigation:** Aviva Levitas, Emad Muhammad, Yuan Zhang, Isaac Perea Gil, Ricardo Serrano, Nashielli Diaz, Maram Arafat, Alexandra A. Gavidia, Yoram Etzion, Ruti Parvari, Ioannis Karakikes.

**Methodology:** Aviva Levitas, Emad Muhammad, Yuan Zhang, Isaac Perea Gil, Nashielli Diaz, Maram Arafat, Yoram Etzion, Ruti Parvari, Ioannis Karakikes.

**Project administration:** Yoram Etzion, Ruti Parvari, Ioannis Karakikes.

**Resources:** Michael S. Kapiloff, Mark Mercola, Yoram Etzion, Ruti Parvari, Ioannis Karakikes.

**Software:** Ricardo Serrano, Mark Mercola.

**Supervision:** Yoram Etzion, Ruti Parvari, Ioannis Karakikes.

**Validation:** Yoram Etzion, Ruti Parvari, Ioannis Karakikes.

**Visualization:** Aviva Levitas, Emad Muhammad, Ricardo Serrano, Ioannis Karakikes.

**Writing – original draft:** Aviva Levitas, Emad Muhammad, Yuan Zhang, Isaac Perea Gil, Ruti Parvari, Ioannis Karakikes.

**Writing – review & editing:** Yuan Zhang, Isaac Perea Gil, Ricardo Serrano, Alexandra A. Gavidia, Michael S. Kapiloff, Ruti Parvari, Ioannis Karakikes.

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
