## [Decision Letter · Decision Letter 0]

28 Apr 2020

Dear Dr Karakikes,

Thank you very much for submitting your Research Article entitled 'A Novel Recessive Mutation in SPEG Causes Early Onset Dilated Cardiomyopathy' to PLOS Genetics. Your manuscript was fully evaluated at the editorial level and by independent peer reviewers. The reviewers appreciated the attention to an important problem, but raised some substantial concerns about the current manuscript. Based on the reviews, we will not be able to accept this version of the manuscript, but we would be willing to review again a much-revised version. We cannot, of course, promise publication at that time.

If you decide to revise the manuscript for further consideration at PLOS Genetics, please aim to resubmit within the next 60 days, unless it will take extra time to address the concerns of the reviewers, in which case we would appreciate an expected resubmission date by email to plosgenetics@plos.org.

[LINK]

We are sorry that we cannot be more positive about your manuscript at this stage. Please do not hesitate to contact us if you have any concerns or questions.

Yours sincerely,

Frank L Conlon

Associate Editor

PLOS Genetics

Scott Williams

Section Editor: Natural Variation

PLOS Genetics

It is recognized in the present environment that the authors may not have access to the lab and therefore, may not be able to conduct the additional requested studies. Therefore, any revision would not require those experiments. However, that said the authors must make every effort to address all other concerns from the reviewers for the manuscript to be considered for publication. Explicitly, the authors must at the least:

1. Include all data not shown. This includes all data requested by the reviewers as well as all data referred to in the manuscript as not shown or not presented.

2. The authors must include all data with their isogenic lines. This is essential.

3. The authors must include what data they have in hand to verify the cells are at the maturity they claim.

4. All text revisions by Reviewer #2.

5. All changes and edits and all response to reviewer’s must be accompanied by changes in the test of the manuscript and not solely in the letter that responds to the reviewer’s comments.

Reviewer's Responses to Questions

**Comments to the Authors:**

Reviewer #1: Dr. Levitas and colleagues propose that a novel mutation in the gene encoding striated muscle enriched protein kinase (SPEG) causes severe autosomal recessive dilated cardiomyopathy in a consanguineous family. They perform linkage analysis to identify SPEG then use CRISPR in human induced pluripotent stem cells to indicate that the SPEG E1680K mutation causes functional deficits consistent with the DCM phenotype. The study is novel, interesting and presented very clearly. A few concerns diminish enthusiasm for the manuscript in its present form.

1.The authors should cascade sequence SPEG in apparently unaffected parents and siblings to confirm the mode of inheritance and offer a sense of penetrance.

2. Were the controls for the hIPSC-CM experiments actually isogenic controls as indicated? hIPSC-CMs with non-targeting gRNA in the vector construct would be more appropriate controls.

3. Details regarding the maturity of the hIPSC-CMs is scant and the WT image in 5c does not look like a cardiomyocyte. How did the authors phenotype and validate their cells to assure maturity?

4. The structural/histological E1680K phenotype in 5c is rather extreme—it is difficult to believe that these cells accurately represent the in vivo milieu. On a potentially related note, why did the authors choose to CRISPR the undifferentiated hIPSCs rather than hIPSC-CMs?

5. It would be relatively straightforward to immunblot for SERCA2a Thr484 phosphorylation to determine whether the E1680K mutation affects SERCA2a function, leading to the abnormalities in calcium handling.

6. Is the difference in diastolic calcium in Figure 5b actually statistically significant (p<0.001)?

7. Two of the affected children had associated congenital heart defects (VSD and ASD). Do the authors consider these defects to be associated with the SPEG mutation?

Reviewer #2: This manuscript provides strong clinical segregation data to support the contention that a recessive variant in SPEG, E1680K, is disease-causing for early onset DCM with childhood lethality in a consanguineous family. The methodologies for linkage analysis, homozygosity mapping and exome sequencing appear to be sound. The iPSC derivation and CM differentiation methodologies are also sound and the phenotype data support a pathogenic role of this recessive variant as there appears to be disruption of sarcomere structure and decreased contractility in unpatterned monolayers and in EHTs.

What is lacking from the manuscript is a mechanism for how this recessive variant reduces contractility and causes DCM. The authors speculate that it has something to do with phosphorylation of SERCA based on the increased Ca transients and faster Ca decay but no data are provided to support this. Additional experimental data, if available, would add significant value. If these data are not available, and not attainable given current widespread laboratory shut downs, then a potential alternative is to add a limitation section that outlines the shortcomings of the current work and the limited functional characterization of these iPS-CM lines.

1. The authors show that SPEG mRNA is not reduced in the iPSC-CMs with recessive E1680K variants. Do they have protein level data by western or mass spec? The variant could destabilize the protein even if the RNA is intact. This would be important to know whether this variant causes complete loss of functional protein vs some other effect on protein function.

2. The claim that SPEG E1680K causes disruption of sarcomere architecture by IF is done in unpatterned cardiac myocytes which is not ideal, and further, not quantified in any objective manner. At the very least, multiple panels of different fields across at least 3 batches of differentiation with some form of semi-quantification should be shown.

3. The higher peak calcium transient and faster calcium decay kinetics are unusual for DCM. Typically in DCM, the SR load is reduced due to decreased activity/density of SERCA which leads to reduced transients. The authors do not measure SR load by caffeine application so it’s difficult to interpret these data. If they have additional calcium data, they should expand their results. If not, they need to speculate as to why these data differ from the vast majority of data on calcium dynamics in DCM and heart failure. Do they have any data on SERCA2a phosphorylation or PLN expression or phosphorylation?

4. A limitations section should be added that emphasizes the lack of mechanism based on the data presented, as well as the fact that homozygous variants are rare in DCM and that SPEG heterozygous variants may not necessarily be disease-causing. They should also soften the final summary paragraph that implies that DCM gene testing panels should be expanded based on this and other similar studies. The clinical segregation data for this example are very strong, but this level of clinical certainty is often not attainable. Functional data can be supportive, but are not sufficient alone to establish pathogenicity of many variants identified in patients.

5. Are there any objective clinical data for skeletal muscle function or injury in any of the patients? (i.e. like CKs?). If not, then they should soften the statement that the affected individuals had no clinical manifestations of myopathy.

6. Which parents do II-2 and 5 come from in the pedigree?

**Have all data underlying the figures and results presented in the manuscript been provided?**

Reviewer #1: Yes

Reviewer #2: None

PLOS authors have the option to publish the peer review history of their article (what does this mean?). If published, this will include your full peer review and any attached files.

Reviewer #1: No

Reviewer #2: Yes: Sharlene M. Day

---

## [Decision Letter · Decision Letter 1]

21 Jul 2020

Dear Dr Karakikes,

We are pleased to inform you that your manuscript entitled "A Novel Recessive Mutation in SPEG Causes Early Onset Dilated Cardiomyopathy" has been editorially accepted for publication in PLOS Genetics. Congratulations!

Yours sincerely,

Frank L Conlon

Associate Editor

PLOS Genetics

Scott Williams

Section Editor: Natural Variation

PLOS Genetics

Comments from the reviewers (if applicable):

Reviewer's Responses to Questions

**Comments to the Authors:**

Reviewer #1: This reviewer appreciates the comprehensive responses to initial critiques and congratulates the authors on their excellent work.

Reviewer #2: The authors have been very responsive to the critiques. My only request is that they add a sentence to the results section regarding their attempt to quantify SPEG protein by western and that it was unsuccessful because of lack of detection by commercially available antibodies.

**Have all data underlying the figures and results presented in the manuscript been provided?**

Reviewer #1: Yes

Reviewer #2: Yes

PLOS authors have the option to publish the peer review history of their article (what does this mean?). If published, this will include your full peer review and any attached files.

Reviewer #1: No

Reviewer #2: **Yes: **Sharlene M. Day

**Data Deposition**

http://datadryad.org/submit?journalID=pgenetics&manu=PGENETICS-D-20-00301R1

**Press Queries**

---

## [Editor Report · Acceptance letter]

9 Sep 2020

PGENETICS-D-20-00301R1 

A Novel Recessive Mutation in SPEG Causes Early Onset Dilated Cardiomyopathy 

Dear Dr Karakikes, 

We are pleased to inform you that your manuscript entitled "A Novel Recessive Mutation in SPEG Causes Early Onset Dilated Cardiomyopathy" has been formally accepted for publication in PLOS Genetics! Your manuscript is now with our production department and you will be notified of the publication date in due course.

With kind regards,

Kaitlin Butler

PLOS Genetics

On behalf of:
